# USP15 Represses Hepatocellular Carcinoma Progression by Regulation of Pathways of Cell Proliferation and Cell Migration: A System Biology Analysis

**DOI:** 10.3390/cancers15051371

**Published:** 2023-02-21

**Authors:** Yiyue Ren, Zhen Song, Jens Rieser, Jörg Ackermann, Ina Koch, Xingyu Lv, Tong Ji, Xiujun Cai

**Affiliations:** 1Department of General Surgery, Sir Run Run Shaw Hospital, School of Medicine and Innovation Center for Minimally Invasive Technique and Device, Zhejiang University, Hangzhou 310016, China; 2Molecular Bioinformatics Group, Institute of Computer Science, Faculty of Computer Science and Mathematics, Goethe University Frankfurt, 60325 Frankfurt am Main, Germany

**Keywords:** USP15, HCC, pathway analysis, system biology, cell behavior

## Abstract

**Simple Summary:**

Expression levels of the protein USP15 correlated with slow proliferation and slow migration of HCC cells. In HCC patients, high expression of USP15 in tumor tissue was associated with significantly decreased risk for mortality and cancer relapse. Overexpression of USP15 led to reduced tumor growth in a mouse model. USP15 regulated indirectly tumor-related proteins through interactions mediated by other proteins. The functions “cell migration” and “cell proliferation” were significantly enriched in a network of pathway hierarchies that we found relevant for the regulatory role of USP15 in HCC. Six cluster of pathways described the link between experimental observed phenotypes and expression levels of USP15.

**Abstract:**

Background: Hepatocellular carcinoma (HCC) leads to 600,000 people’s deaths every year. The protein ubiquitin carboxyl-terminal hydrolase 15 (USP15) is a ubiquitin-specific protease. The role of USP15 in HCC is still unclear. Method: We studied the function of USP15 in HCC from the viewpoint of systems biology and investigated possible implications using experimental methods, such as real-time polymerase chain reaction (qPCR), Western blotting, clustered regularly interspaced short palindromic repeats (CRISPR), and next-generation sequencing (NGS). We investigated tissues samples of 102 patients who underwent liver resection between January 2006 and December 2010 at the Sir Run Run Shaw Hospital (SRRSH). Tissue samples were immunochemically stained; a trained pathologist then scored the tissue by visual inspection, and we compared the survival data of two groups of patients by means of Kaplan–Meier curves. We applied assays for cell migration, cell growth, and wound healing. We studied tumor formation in a mouse model. Results: HCC patients (*n* = 26) with high expression of USP15 had a higher survival rate than patients (*n* = 76) with low expression. We confirmed a suppressive role of USP15 in HCC using in vitro and in vivo tests. Based on publicly available data, we constructed a PPI network in which 143 genes were related to USP15 (HCC genes). We combined the 143 HCC genes with results of an experimental investigation to identify 225 pathways that may be related simultaneously to USP15 and HCC (tumor pathways). We found the 225 pathways enriched in the functional groups of cell proliferation and cell migration. The 225 pathways determined six clusters of pathways in which terms such as signal transduction, cell cycle, gene expression, and DNA repair related the expression of USP15 to tumorigenesis. Conclusion: USP15 may suppress tumorigenesis of HCC by regulating pathway clusters of signal transduction for gene expression, cell cycle, and DNA repair. For the first time, the tumorigenesis of HCC is studied from the viewpoint of the pathway cluster.

## 1. Introduction

Hepatocellular carcinoma (HCC) is one of the most common cancer types worldwide. The development of treatment methods and research on relevant mechanisms is an important topic in current cancer research [1]. HCC accounts for the majority of cases of primary liver cancer, and every year, more than 600,000 people die of HCC [2,3]. In the focus of current molecular cancer therapies have been protein kinases that mediate antitumor activity primarily through abnormal expression and tumor-dependent or unregulated inactivation of target enzymes. Efforts have increased the number of treatment options [4]. Sorafenib (FDA 2007) and Lenvatinib (FDA 2018) are clinical first-line drugs in the treatment strategy of liver cancer. 

Post-translational modification, such as ubiquitination, plays an important role in tumorigenesis [5,6,7,8]. Ubiquitination is related to proteasome degradation [9]. Tagging genes such as p27, p53, or NF-κB, ubiquitin (Ub) influences pathways that play an important role in tumorigenesis [5,6,7,8]. Ubiquitin-specific proteases (USPs) remove ubiquitin from proteins and hence may regulate tumor-related pathways [10,11]. Its simple chemical structure makes ubiquitin a preferable target for anticancer therapy [12]. The adjustment of the balance between ubiquitination and deubiquitination may lead to lesser side effects than targeting the large proteasome complex with drugs such as Bortezomib [13]. 

Ubiquitin carboxyl-terminal hydrolase 15 (USP15) is a member of the family of ubiquitin-specific proteases. USP15 shows high similarity with the proto-oncogene USP4, which cleaves ubiquitin [14]. USP15 was identified in 1999 and its role in tumorigenesis has been reported recently. A relationship between USP15 and cancer has been proposed by the observation of its stabilization function on the TGF-β receptor in glioblastoma [15]. USP15 has been shown to be elevated in breast and ovarian cancer [15]. USP15 has been shown to repress apoptosis in melanoma and colorectal cancer cell lines through stabilizing the E3 ubiquitin protein ligase MDM2 [16]. USP15 has been shown to suppress glioblastoma cell growth through stabilizing the E3 ubiquitin protein ligase HECTD1 [17]. USP15 has been shown to regulate DNA homologous recombination repair, which has been related to tumorigenesis in the breast cancer cell line MCF7 and human osteosarcoma cell line U2OS [18]. USP15 may be expected to regulate ubiquitination pathways of proteins such as TGF-β, MDM2-p53, Keap1/Nrf2, and Wnt/β-catenin. Ubiquitination pathways may play an important role in HCC. USP15 may regulate tumorigenesis in HCC by multiple pathways. The role of USP15 in HCC is, however, unknown. USP15 has been shown to act as an oncogene or as a tumor-suppressing gene [15,16,17,18]. Systems biology applies network analysis and data integration to explore biomedicine questions. The holistic view of systems biology may help to unravel the complex function of USP15 in the progression of HCC. We applied algorithms of systems biology and experimental methods to investigate the function and mechanism of USP15 in HCC. 

## 2. Materials and Methods 

### 2.1. Tumor-Normal Comparison of Publicly Available Data of GTEx/TCGA

We performed a tumor-normal comparison with the database of UCSC Xena (https://xena.ucsc.edu/#analysis, accessed on 5 May 2022) [19] and the tool Gene Expression Profiling Interactive Analysis 2 (GEPIA2, http://gepia2.cancer-pku.cn/#index, accessed on 7 March 2020) [20]. The Xena provided data sources of the Genotype-Tissue Expression Project (GTEx) of The Cancer Genome Atlas (TCGA). We chose the publicly available dataset of liver hepatocellular carcinoma (LIHC) with 369 primary tumor tissue samples (no recurrent tumor) from TCGA and 160 samples from GTEx (dataset: TCGA TARGET GTEx). Boxplot analysis for USP15 expression of RNA-Seq by Expectation–Maximization (RESM) was applied in log scale (dataset: gene expression RNAseq-RSEM norm_count), see Figure 1A. With GEPIA2, we considered mRNA levels of USP15. We split the 369 tissue samples (no recurrent tumor) into two equal-sized groups; one with high expression of USP15 and a second group with low expression of USP15. Before splitting, we normalized the mRNA level of each tissue to the corresponding mRNA level of the control glyceraldehyde 3-phosphate dehydrogenase (GADPH). We used the median value of the 369 normalized USP levels as the threshold for the split between the two groups. We compared the Kaplan–Meier plots of the two groups, see Figure 1B. To compute the hazard ratio, we applied the Cox Proportional-Hazards Model. We chose a 95% confidence interval (CI). We applied the log-rank test to compare two Kaplan–Meier curves. 

### 2.2. Patient Cohort and Tissue Samples

We investigated tissues samples of 102 patients that were diagnosed with HCC and underwent liver resection between January 2006 and December 2010 at the Sir Run Run Shaw Hospital (SRRSH). For detailed information on the clinical and pathological characteristics of the patient cohort, we refer to Table 1. We collected a cancerous tissue sample for each of the 102 patient samples. As a control, we took 27 samples from non-cancerous tissue adjacent to the tumor. All specimens were immediately immobilized in formalin and embedded in paraffin block for subsequent pathological testing. We used ten pairs of fresh frozen tissue specimens from the hospital tissue bank of SRRSH for Western blot analysis. We evaluated tumor stages according to the American Joint Committee on Cancer (AJCC) staging system and defined clinical stages according to the Barcelona Clinic Liver Cancer (BCLC) staging system. This study was approved by the Medical Ethics Committee of SRRSH with the number: 20190211-90, and all patients were informed. All the procedures conformed with the Declaration of Helsinki [21]. 

### 2.3. Cell Culture

Cell lines HCCLM3, SKhep-1, HA22T, HepG2, Huh1, Huh7, hep3B, QGY-7703, SMMC-7721, Bel-7404, SNU-387, SNU-449, SNU-423, and SNU-745 were provided by the Cang laboratory at Zhejiang University, Hangzhou, China. Cell lines HCCLM3, SKhep-1, HA22T, HepG2, Huh1, Huh7, hep3B, SMMC-7721, Bel-7404, and LO-2 were grown in DMEM (Gibco, Waltham, MA, USA, cat#C11995500BT) supplemented with 10% fetal bovine serum (FBS) (Cellmax, Beijing, China cat#SA102.02). Cell lines SNU-387, SNU-449, SNU-423, SNU-745, and QGY-7703 were grown in RPMI-1640 (Gibco, cat# C11975500BT) supplemented with 10% FBS (Cellmax, cat#SA102.02). All cells were maintained in an atmosphere of 5% CO_2_ in a humidified 37 °C incubator. 

### 2.4. Plasmid Constructs

We designed the single-guide RNA (sgRNA), using the online sgRNA design website (http://crispr-era.stanford.edu/, accessed on 12 November 2017). The sgRNA was cloned into the lentiCRISPRv2 plasmid (Addgene, Watertown, MA, USA cat#52961, standard protocol). Full-length USP15 cDNA was kindly provided by the Cang laboratory (Zhejiang University). The cDNA was amplified and cloned into the expression vector p3xFlag-CMV-7.1 (Sigma, St. Louis, MO, USA, cat#E4026) using DNA restriction enzymes XbaI and BamHI. We cloned the full-length cDNA of USP15 into the plasmid pXF4H [23] at the restriction sites EcoRI and BamHI; pXF4H was provided by the Cang laboratory. For the sequences of the primers, we refer to Table 2.

### 2.5. Lentivirus Production and Transduction

Lentivirus expressing sgRNA was produced by co-transfecting cells HEK293T with a mixture of plasmid lentiCRISPRv2-sgUSP15 (or plasmid lentiCRISPRv2-sgControl), the Gag-Pol packaging plasmid psPAX2 (Addgene, cat#12260), and the envelope plasmid B19/VSVG (Addgene, cat#88865) at a 4:3:2 ratio. We used transfection reagent Lipofectamine 2000 (lipo2000, Thermo Fisher Scientific, Waltham, MA, USA, cat#11668019). Virus particles were harvested 48 h after transfection. Lentivirus-infected cells were screened with puromycin (Thermo Fisher Scientific, cat#A1113802: Huh1,1µg/µL; HA22T, 1µg/µL).

### 2.6. Quantitative Real-Time PCR Analysis

RNA was purified from liver tissue samples using Trizol (Thermo Fisher Scientific cat#204211) according to the manufacturer’s protocol. The reverse transcript was performed using PrimeScript RT Master Mix kit (Takara, Tokyo, Japan, cat#RR037B). Real-time PCR was carried out in SYBR Green PCR Master Mix (Bio-Rad, Hercules, CA, USA, cat#1725270) with ABI PRISM 7500 Sequence Detection System. The USP15 primer sequences were as follows: forward, 5’-CTG CTC AAA ACC TCG CTC C-3’ and reverse, 5’-CAA TGG GTC CAG GAT ACA CA-3’. The GAPDH primer sequences were as follows: forward, 5’-AGGTCGGTGTGAACGGATTTG-3’ and reverse, 5’-TGTAGACCATGTAGTTGAGGTCA-3’. Each measurement was performed in triplicate, and results were normalized to the expression of the internal reference gene GAPDH.

### 2.7. Western Blot and Antibodies

For protein extraction, tumor tissues or cell pellets were homogenized in the buffer of RIPA (Sigma, cat#R0278), which contained a cocktail of protease inhibitors (Roche, Basel, Switzerland, #4693116001). After incubation on ice for 30 minutes, we centrifuged for 15 minutes at 4 °C at 12,000× *g*. Protein concentration was quantified using the kit BCA protein assay (Thermo Fisher Scientific, cat#A53227). Equal amounts of protein were separated discontinuously on a 4–12% SDS–PAGE gel and transferred to polyvinylidene difluoride (PVDF) membrane (Merck Millipore, Billerica, MA, USA). Non-specific binding sites on the membranes were blocked for 1 h with 5% non-fat milk (BD, Franklin Lakes, NJ, USA, cat#232100). After blocking, membranes were incubated overnight at 4 °C with primary antibodies, washed three times with TBS/T, and incubated with HRP-conjugated secondary antibodies for 1 h at room temperature. Immunoreactions were visualized using Clarity Western ECL substrate (Bio-Rad, cat#VL001), and the blots were imaged using a luminescent image analyzer (Fujifilm, Tokyo, Japan). The following antibodies were used: anti-USP15 (Santa Cruz, Santa Cruz, CA, USA, cat#sc-100629), anti-β-catenin (Cell Signaling Technology, Danvers, MA, USA, cat#8480), anti-Phospho-Nrf2 (Ser40) (Thermo Fisher Scientific, cat#PA5-67520), anti-Phospho-Rb1 (Ser780) (Cell Signaling Technology, cat#9307), anti-IκBα (Cell Signaling Technology, cat#4814), anti-N-cadherin (Cell Signaling Technology, cat#13116), anti-Phospho-Smad2 (Ser465/467) (Cell Signaling Technology, cat#3108), anti-Phospho-p44/42 MAPK (Erk1/2) (Thr202/Tyr204) (Cell Signaling Technology, cat#4370), anti-Phospho-AKT (Ser473) (Cell Signaling Technology, cat#4060), anti-C-myc (Cell Signaling Technology, cat#18583), anti-SQSTM1 (P62) (Abcam, Cambridge, UK, cat#ab56416), anti-β-actin (Cell Signaling Technology, cat#3700), and anti-GAPDH (Abcam, cat#ab181602).

### 2.8. Immunohistochemical Staining

Paraffin-embedded sections were provided by the department of pathology of the SRRSH. The tissues were paraffin-embedded using a Tissue-Tek automated processor (Sakura) and cut into 3 μm tissue sections. The tissue sections were deparaffinized and rehydrated. For antigen retrieval, the sections were immersed in 10 mM citrate buffer (pH 6.0) and boiled for 10 min in a microwave oven. Endogenous peroxidase activity was blocked with 3% hydrogen peroxide for 10 min. Non-specific binding sites were blocked with 5% normal goat serum for 30 min. The sections were incubated with an antibody against USP15 (1:100, Santa Cruz, cat#sc-100629) overnight at 4 °C. The sections were incubated with the secondary antibody, and the expression of USP15 in the tissues was observed via microscopy after DAB staining and hematoxylin staining. 

### 2.9. Tissue Scoring

By visual inspection, two trained pathologists assigned a score—strong, moderate, weak, or negative—to an immunochemically stained tissue. All decisions were made blinded to the clinicopathologic characteristics. Appendix A shows a reference image for each of four staining scores. If the decision was not unanimous, a third pathologist repeated the evaluation and made an independent decision. We denoted tissue with the scores strong or moderate as tissue with high expression of USP15. We denoted tissue with the scores weak or negative as tissue with low expression of USP15. For the comparison of the scores of two patient groups, we replaced the scores negative, weak, moderate, and strong by the numbers 1, 2, 3, and 4, respectively, and applied the Mann–Whitney U test. All tissue slides are accessible on reasonable request.

### 2.10. Cell Migration Assay

The cell lines Huh1 and HA22T cells with USP15-stable overexpression and/or knockout (200 μL FBS-free DMEM medium suspension, 1 × 10^5^/mlHA22T or HUH1) were sucked into the upper chamber of Transwell (PC membrane with 8.0-μm pore size). After 24 h of incubation, the chambers were washed with PBS (pH 7.4) three times to remove the cells in the upper chamber. The cells were fixed with 4% paraformaldehyde for 15 min then stained with crystal violet (0.01% in ethanol) for 25 min followed by washing three times. The cells were counted in an inverted microscope, and five visions were randomly taken at ×200. The average number of cells was considered.

### 2.11. Wound Healing Assay

Wound healing assay was carried out by scratching a 6-well dish with a 10-μL pipette tip when the dish was at 80% confluences (including SMMC-7721vec/usp15 overexpression and QGY-7703vec/usp15 overexpression). The widths of the scratches were evaluated at time points 0, 6, 12, 24, and 30 h after scratching.

### 2.12. Cell Growth Assay

We seeded in round numbers 1 × 10^3^ cells per well into 96-well plates. We added cell counting kit CCK8 (5 mg/ml, YEASEN, cat#40203ES60) to the wells. We measured absorbance at wavelength 450 nm. 

### 2.13. Tumor Formation in Nude Mice

BALB/c nude mice (*n* = 10, 4 weeks of age, female) were obtained from ZhiYuan biological company (Hangzhou, China). The nude mice were housed in a pathogen-free facility under controlled temperature (22 ± 2 °C) and lighting (12-h light/dark cycle) conditions. 

We used two human liver cell lines, Huh7-PLVX and Huh7-PLVX-USP15. Huh7-PLVX was the Huh7 cell line transfected with an empty lentiviral expression vector (PLVX). We used cell line Huh7-PLVX as a control. The second cell line, Huh7-PLVX-USP15, was the Huh7 cell line transfected with a lentiviral expression vector to overexpress USP15. We compared the two cell lines Huh7-PLVX-USP15 and cell line Huh7-PLVX to evaluate the effect of high expression of USP15.

Both of the cell lines were suspended in sterilized PBS. Mice were randomized and subcutaneously injected with either cell line Huh7-PLVX or cell line Huh7-PLVX-USP15 (five mice per group, per mouse: 5 × 10^6^ cells suspended in 100 µL of PBS,). After 25 days, the mice were euthanized, and the tumors were excised. The dimensions of the implanted tumors were measured with a vernier caliper. The tumor volume was computed by the formula: volume (cm^3^) = 0.5 × length (cm) × width^2^ (cm^2^). For further analysis, e.g., immunochemical staining, the tumors were embedded in paraffin. All animal experiments were performed under the guidelines reviewed by the Animal Ethics Committee of the Biological Resource Centre of the Agency for Science, Technology and Research at the Sir Run Run Shaw Hospital.

### 2.14. Protein–Protein Interaction Network

Interaction partners of USP15 (layer 1 proteins) were searched in the IntAct database (https://www.ebi.ac.uk/intact/, accessed on 15 March 2020) [24] with species restriction to human. We found 106 interaction partners including USP15 (layer 1 proteins). We applied a breadth-first search (BFS) strategy to expand the set of layer 1 proteins. We found 6175 interactions partners (layer 2 proteins) for the layer 1 proteins (IntaAct, human). For visualization of interaction networks, we chose the circle layout of Cytoscape 3.8 [25]. 

We constructed sub-networks to describe the interaction between USP15 and given small sets of proteins, called target genes. The sub-network had a node for USP15, each target gene, and each of their interaction partners. Each edges denoted an interaction reported in IntAct for human. 

### 2.15. HCC Oncology Gene Collection and USP15 Cancer Gene List

We downloaded the list of genes of HCC from the cBioPortal for Cancer Genomics (Dataset: HCC, TCGA, PanCancer Atlas) [26,27]. We selected genes with a mutation rate > 1%. The HCCgenes were checked to be registered in the OncoKB database [24], see Appendix A for a list of the 295 identified genes. We determined a set of 143 genes that simultaneously were gene of HCC and interaction partners of USP15 (layer 1 protein or layer 2 protein). We denoted the 143 genes that interact with USP15 as the set of HCC-USP15 genes. For a complete list, we refer to Appendix A. 

### 2.16. Next-Generation Sequencing

The human hepatoma cell lines HA22T sg-control and knockout HA22T sgUSP15 were treated with 1 mL of TRIzol reagent (Thermo Fisher Scientific, cat#204211) per 10^7^ cells and stored at −80 °C and afterwards sent in three replicates to Hangzhou Kaitai Biotechnology Co., Ltd. (Hangzhou, China) to perform mRNA extraction and sequencing. The tool DESeq2 was applied to estimate log-fold changes based on mean counts of triple replicates and false discovery rates (FDR) [28]. The thresholds for the selection of significant genes were set to FDR < 0.1 and log2 fold changes Log2FC > 1.

### 2.17. Network of Pathway Hierarchies and Mapping of Functions 

We downloaded the relation of pathway hierarchies from Reactome (https://reactome.org/download/current/ReactomePathwaysRelation.txt, accessed on 20 July 2020) [29]. Only human pathways were considered. A network represented each pathway hierarchy as a node and each relation between two pathway hierarchies as an arc. The network had 2423 nodes and 2424 directed edges. We imported the network in Cytoscape 3.8 [25,29,30]. 

For the mapping, we extracted a list of Uniprot IDs for the genes from the database Uniprot (https://www.uniprot.org/uploadlists/, accessed on 28 July 2020) [31] and downloaded the mapping list “UniProt2Reactome_All_Levels” from Reactome (https://reactome.org/download/current/UniProt2Reactome.txt, accessed on 28 July 2020) [29]. We selected only the human entries in the mapping list to generate the file “UniProt2Reactome_human.txt”. We filtered the Uniprot IDs of the genes in the human mapping list and generated a relevant mapping list “pathway_matching_all.txt”. We imported the file “pathway_matching_all.txt” into the network of pathway hierarchies in Cytoscape. For the enrichment analysis, we mapped all the genes extracted from the file “UniProt2Reactome_All_Levels.txt” to the network of pathway hierarchies, applying our Python script for computing the fraction of genes in each pathway.

### 2.18. Sub-Networks of Pathway Hierarchies 

The complete network of human pathway hierarchies was analyzed using the tool CentiScape [32]. We identified 28 source nodes with an in-degree of zero to decompose the network into the 28 sub-networks of its connected components. Each sub-network was associated with an individual source node. The sizes of the sub-networks ranged from 606 nodes for the hierarchy “diseases” to only 4 nodes for the hierarchy “circadian clock”.

### 2.19. Statistical Analysis

For statistical testing, we applied the tools GraphPad Prism 7 (San Diego, CA, USA), SPSS (V25.0, Chicago, IL, USA), and Microsoft Excel (Office 2021, Redmond, WA, USA). If not indicated otherwise, we give numbers in the form of mean ± standard error. We applied Fisher’s exact test for analysis of contingency tables, i.e., to evaluate possible biases in the patient cohort, see Table 1. We applied the Mann–Whitney U test to compare the USP15 levels in tumor and non-tumor tissue in 102 patients. To compare the survival distribution of groups of patients, we produced Kaplan–Meier plots and applied the log-rank test (Mantel–Cox test). For enrichment analysis, we applied the hypergeometric test. We denoted p-values *p* <0.05 as significant. In graphical representation, we represented significance values as follows: * for *p* < 0.05, ** for *p* < 0.01, *** for *p* < 0.001, and **** for *p* < 0.0001. 

## 3. Results

### 3.1. Expression of USP15 Was Low in Tumor Tissue; High Expression of USP15 Correlated with Decreased Risk for Mortality and Cancer Relapse 

We inspected publicly available data in the Xena database and gene expression profiling interactive analysis tool (GEPIA2) [33]. We chose the publicly available dataset of liver hepatocellular carcinoma (LIHC) with 369 tissue samples from TCGA and 160 samples from GTEx for tumor. The USP15 expression levels in tissue samples of liver cancer were significantly lower than those in normal tissue (*p* < 0.0001, Mann–Whitney U test, Figure 1A). We split the patients into two equal-sized groups, one with high expression of USP15 and a second group with low expression of USP15. The Kaplan–Meier plot of the survival data demonstrated a significantly higher survival rate for patients (expression levels above median) with high expression of USP15 than for patients (expression levels below median) with low expression (Figure 1B, *p* = 0.018, log-rank test). 

**Figure 1 cancers-15-01371-f001:**
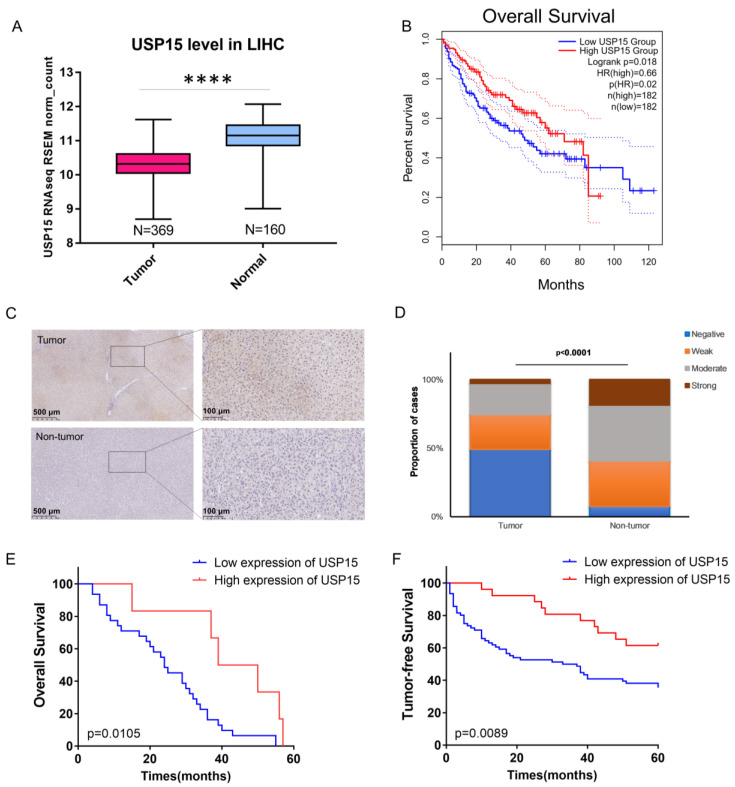
USP15 is down-regulated in hepatocellular carcinoma (HCC) tissue. High expression of USP15 in HCC patients is associated with significantly decreased risk for mortality and cancer relapse. (**A**) Box plots of USP15 expression in tumor and non-tumor tissue; liver hepatocellular carcinoma (LIHC) dataset of TCGA and GTEX. The median expression level of USP15 is lower in tumor tissue (*N* = 369) than in non-tumor tissue (*N* = 160). (****: *p* < 0.0001, Mann–Whitney U test). (**B**) Kaplan–Meier plots of patients in the LIHC dataset of TCGA with high expression of USP15 (red curve) and low expression of USP15 (blue curve). The dotted lines indicate a 95% confidence interval. High expression of USP15 is associated with significantly decreased risk for mortality: hazard ratio 0.6, *p*-value 0.018 (log-rank test). (**C**) Two exemplary images of tissue samples that were stained (IHC) for USP15. Right figures are blowups of a part of the left image, see rectangle. By visual inspection, two trained pathologists assigned tumor tissue samples of 102 patient to one of the two classes: high expression of USP15 (*n* = 26, moderate or strong staining) and low expression of USP15 (*n* = 76, negative or weak staining). T—tumor tissues; NT—para-HCC tissues. (**D**) Fractions of tissue samples that were assigned to four staining levels: negative, weak, moderate, and strong. Tumor (*n* = 102) and non-tumor samples (*n* = 27) have significantly different distributions (*p* < 0.001, Mann–Whitney U test). (**E**) Kaplan–Meier plots for survival of patients with low expression (*n* = 76, blue curve) and high expression (*n* = 26, red curve). High expression of USP15 is associated with significantly decreased risk for mortality: hazard ratio 0.379, *p* = 0.0105 (log-rank test). (**F**) Kaplan–Meier plots for tumor-free survival of patients with low expression (*n* = 76, blue curve) and high expression (*n* = 26, red curve). High expression of USP15 is associated with significantly decreased risk for cancer relapse: hazard ratio 0.420, *p* = 0.0089 (log-rank test).

We retrospectively inspected tissue samples from 102 patients with a diagnosis of HCC at Sir Run Run Shaw Hospital (SRRSH). Information on the cohort of patients is listed in Table 1. Tumor and non-tumor tissue samples were immunochemically stained for USP15 (Figure 1C). By visual inspection, we classified tissue samples according to the degree of staining into four levels: negative, weak, moderate, and strong (Appendix A). A total of 59.2% of non-tumor tissue samples were assigned to strong or moderate staining. In contrast, only 25.4% of tumor samples were assigned to strong or moderate staining. (Appendix A). The staining of tissue samples differed significantly in tumor samples and non-tumor samples (*p* < 0.0001, Mann–Whitney U test, Figure 1D). In accordance with our findings for the publicly available dataset LIHC, USP15 expression tended to be reduced in the tumor tissues of our patient cohort. 

We split the 102 patients in two groups. Patients with tumor tissue of negative or weak staining were assigned to the group low expression of USP15 (*n* = 76). Patients with tumor tissue of moderate or strong staining were assigned to a second group, high expression of USP15 (*n* = 26). We compared the survival data of the two groups by means of Kaplan–Meier plots. High expression of USP15 was associated with significantly decreased risk for mortality: hazard ratio 0.379, *p* = 0.0105 (log-rank test, Figure 1E). We compared data of tumor-free time periods by means of Kaplan–Meier plots. High expression of USP15 was associated with significantly decreased risk for cancer relapse: hazard ratio 0.420, *p* = 0.0089 (log-rank test, Figure 1F). The correlation of high survival rate and reduced cancer relapse with high expression of USP15 might suggest a role of USP15 as a tumor suppressor. 

### 3.2. USP15 Expression Was Increased in HCC Cell Lines 

We compared the protein level of USP15 in the HCC cell lines Huh1, HA22T, and Huh7 with the protein level of USP15 in cell lines LM3, PLC/PRF/5, HLF, and Hep-G2. To measure the level of USP15, we applied Western blotting. For each cell line, we normalized the luminescence of a USP15 antibody by luminescence of a GAPDH antibody. We found the highest normalized expression levels of USP15 for HCC cell lines Huh1, HA22T, and Huh7 (Figure 2A,B; the uncropped Western blots are shown in Appendix A). This finding supported our decision to choose HA22T and Huh1 cell lines to be used as a cell model for researching USP15.

### 3.3. Overexpressing USP15 Correlated with Slow Proliferation and Slow Migration 

We transfected USP15 to cell lines Huh1 and HA22T to overexpress USP15. Overexpressing USP15 in cells Huh1 (Huh1-P-F-USP15) had a reduced proliferation index of, in round numbers, 50% of the control (*p* values at five time points all less than 0.0001, two-tailed *t*-test, Figure 2C). Overexpressing USP15 in cells HA22T (HA22T-P-F-USP15) had a slightly decreased proliferation index compared to the control and the proliferation was reduced only 2.86% (p values at 48 h, 60 h and 72 h are less than 0.05, two-tailed *t*-test, Figure 2D). We applied a migration assay (Transwell) to investigate the motility of the overexpressing cell lines. USP15 overexpressing cells Huh1 (Huh1-P-F-USP15) had a reduced migration rate of 45% of the migration rate of the control (*p* < 0.01, two-tailed *t*-test, Figure 2E,F). On USP15 overexpressing cells HA22T (HA22T-P-F-USP15), the migration rate was nearly completely inhibited compared to the control c (*p* < 0.0001, two-tailed *t*-test, Figure 2G,H). 

We transfected cell lines QGY-7703 and SMMC-7721 to overexpress USP15. We applied wound healing assays to evaluate the effect of overexpression of USP15. Overexpressing cells, QGY-7703-PXF4H-USP15 and SMMC-7721-PXF4H-USP15, had a lower movement rate than their controls (two-tailed *t*-test, Appendix A). The USP15 overexpression efficiency in the two cell lines is shown in Appendix A. 

We also use flow cytometry to detect whether the level of USP15 influences the apoptosis rate of the cell. However, in Huh1 and HA22T cell lines, overexpression or knockout USP15 does not change the apoptosis rate of the two cell lines (Appendix A), which indicated that a high level of USP15 represses the proliferation of these cell lines.

### 3.4. Knockout of USP15 Correlated with High Proliferation and High Migration 

We tested whether knockout had reverse effects compared to overexpression of USP15. Hepatocellular cell lines Huh1 and HA22T were modified with CRISPR/Cas9 to knock out expression of USP15. The reduced expression of USP15 was tested for knockout cell lines Huh1-sgUSP15 and HA22T-sgUSP15 using Western blotting (Figure 2I,M). Increased proliferation index was observed for knockout cell lines Huh1-sgUSP15 Figure 2J) and HA22T-sgUSP15 (*p* value at 72 h less than 0.05, two-tailed *t*-test, Figure 2N). Migration assays showed increased motility knockout cell lines Huh1-sgUSP15 (*p* < 0.01, two-tailed *t*-test, Figure 2K,L) and HA22T-sgUSP15 (*p* < 0.01, two-tailed *t*-test, Figure 2O,P).

### 3.5. Overexpression of USP15 Correlated with Slow Tumor Growth in Mice

To investigate a role of USP15 as a possible tumor suppressor, we subcutaneously injected Huh7 cells into BALB/c nude mice. Five mice, the control group, were injected with cells of control cell line, Huh7-PLVX. Huh7-PLVX were Huh7 cells that were transfected by an empty vector. A second group of five mice, OE group, were injected with cells of cell line Huh7-PLVX-USP15. Huh7-PLVX-USP15 were Huh7 cells that were transfected with a lentiviral expression vector to overexpress USP15. Tumors were excised after 25 days (Figure 2Q). We confirmed a high expression of USP15 by immunostaining of tumor tissue (Figure 2R). We found the volume of tumor significantly larger in the OE group (*p* = 0.0079, two-tailed *t*-test) (Figure 2S), and the weight of tumor was also significantly larger in the OE group (two-tailed *t*-test) (Figure 2T).

### 3.6. Tumor-Related Proteins Were Indirectly Regulated by USP15 via Interactions Mediated by Other Proteins 

We selected the proteins from the IntAct database [34] that interact with USP15 (layer 1 proteins). To the set of 105 interaction partners of USP15, we added their interaction partners (layer 2) (Figure 3A). For the set of proteins, we constructed the protein–protein interaction (PPI) network. We deleted all non-human genes and compounds. The resulting network had 105 proteins in layer 1 (Figure 3B) and 6175 proteins in layer 2 (Figure 3C). The PPI network had 17,439 edges. In the following, we denote the network as the human USP15 PPI network.

With the searching method shown in Figure 3D, we chose a set of the ten genes (Figure 3E) to see the relations of them to USP15. The ten genes using two conditions: (1) the gene was known to be tumor-related and (2) the availability of an antibody enabled an experimental investigation by Western blotting in our lab. Eight of the ten proteins were members of the human USP15 PPI network (Figure 3F,G). We applied Western blotting to compare gene expression and phosphorylation levels in cell lines Huh1-P-F_USP15 (OE, overexpressing USP15) and Huh1-sgUSP15 (KO, knockout USP15) versus their control cell lines Huh1-PLVX and Huh1-sgControl, respectively, see Material and Methods. We assumed proteins to be regulated by USP15 if overexpression and knockout had reverse effects. We found the eight members of the human USP15 PPI network regulated by USP15 (Figure 4A–I). Also regulated by USP15 was Nrf2, which was not part of the human USP15 PPI network. Nrf2 was, however, an interaction partner of layer 2 proteins. We also tested the level of some downstream proteins of USP15 in the tumor samples from the mouse model. The level of beta-catenin increased, whereas pAKT and pERK decreased (Figure 4J,K). These proteins level changes revealed the mechanism by which USP15 represses tumor growth.

### 3.7. Regulatory Role of USP15 in HCC: Relevant Pathway Hierarchies 

We constructed a network of all human pathway hierarchies, see Material and Methods. Nodes and arcs represented pathway hierarchies and hierarchy relations, respectively. The initial network had 2423 nodes and 2424 arcs. 

Motivated by the experimental evidence of a regulation by USP15 for eight exemplary members of the human USP15 PPI network, we chose all liver cancer genes with a mutation rate of over 1% (295 genes, Appendix A) in the database cBioPortal for Cancer Genomics (dataset: HCC, TCGA, PanCancer Atlas) [26,27]. We identified 143 of the liver cancer genes as members of the human USP PPI network (HCC-USP15 genes, Appendix A, Figure 5A). We mapped the 143 HCC-USP15 genes to the network of human pathway hierarchies. The number of HCC-USP15 genes per node varied between $n_{HCC}=1$ and $n_{HCC}=91$ (median 2, mean 3.56 ± 2.82).

Using NGS, we measured log-fold change in gene expression after a knockout of USP15 in cell line HA22T (Figure 5B). We found 1.4% (*n* = 203) and 2.0% (*n* = 278) of the 14,099 identified proteins up-regulated and down-regulated, respectively. We mapped the set of 481 differentially expressed genes to the network of human pathway hierarchies. The number of differentially expressed genes per node varied between $n_{diff}=1$ and $n_{diff}=79$ (median, 1; mean, 2.55 ± 1.87).

To compute an upper bound for the number of genes, we mapped a list of all human genes to the network of human pathway hierarchies. The total number of genes that may be mapped to a node varied between $n_{diff}=1$ and $n_{diff}=3443\;$ (median, 17; mean, 56.37 ± 63.63). We ranked the nodes by the score: $$s=\frac{n_{diff}\;n_{HCC}}{n_{tot}^{2}{}_{}}\;\;10^{4\;}.$$

A pathway hierarchy with all associated genes being differentially expressed and simultaneously, being an HCC-USP15 gene, e.g., with $n_{diff}=n_{HCC}=n_{tot}$, would reach the theoretical upper bound of *s* = 10^4^. The score, *s*, was intended to quantify the specificity of a node for the 624 genes that may contribute to the regulatory role of USP15 in HCC. The values of *s* varied between 0 and 625 (median, 0; mean, 7.91 ± 8.49; Appendix A). For a color-coded representation of the scores, we refer to Figure 5C, and Appendix A is the original network which is used as the negative control. We selected a subset of 225 nodes (9.2%) with score values above s = 20. Note that a node with a small fraction of differentially expressed genes and a small fraction of HCC genes, such as $\;n_{diff}/n_{tot}=\;n_{HCC}/n_{tot}=4.4\%\;$, would get a score below 20. Appendix A was used as blank figure.

We denoted the sub-network for the 225 “high score” pathway hierarchies as HCC-USP15 pathways (Figure 6A).

### 3.8. Regulatory Roles of USP15 in HCC Were Cell Proliferation and Migration 

We downloaded 163, 75, and 143 pathway hierarchies (database: Reactome) that were associated with cell proliferation, cell migration, and lipid processes, respectively. We tested the hypothesis of a random selection of pathway hierarchies for network HCC-USP15 pathways. Since HCC-USP15 pathways had 225 nodes, we would expect to find mean numbers of n=163×225/2423≈15.14, n=75×225/2423≈6.96 , and n=143×225/2423≈13.28  pathway hierarchies associated with cell proliferation, cell migration, and lipid processes, respectively. 

In the network HCC-USP15 pathways, we found an enrichment of 35 pathways (expected mean 15.14, Appendix A-proliferation) associated with cell proliferation (*p* = 0.0077, hypergeometric test). For a color-coded visualization of pathway hierarchies associated with cell proliferation, we refer to Figure 6B. We found an enrichment of 13 pathways (expected mean 6.96 Appendix A-migration) associated with cell migration (*p* = 2.5 × 10^−7^, hypergeometric test). For a color-coded visualization of pathway hierarchies associated with cell proliferation, we refer to Figure 6C. We found no enrichment of pathways associated with lipid processes. Only seven pathways (expected mean 13.28, Appendix A-lipid processes) were associated with lipid processes (*p* = 0.035, hypergeometric test). For a color-coded visualization of pathway hierarchies associated with lipid processes, we refer to Figure 6D. 

The complete network of human pathway hierarchies clustered into 28 connected components, so-called clusters (Table 3). We mapped the 225 “high score” pathway hierarchies, i.e., HCC-USP15 pathways, to the 28 clusters. HCC-USP15 pathways were significantly enriched (hypergeometric test) in 8 of the 28 pathway clusters. We assigned six of the enriched clusters to cancer pathways (Figure 7A). For a discussion of cancer-related pathways, we refer to Hanahan and Weinberg [35]. The six tumor-related clusters were “Cellular responses to external stimuli” (33.33%, 0.000165321, Figure 7B), “Cell Cycle” (19.20%, *p* = 0.000122438, Figure 7C), “DNA repair” (34.43%, *p* = 6.07738E-09, Figure 7D), “Gene expression” (14.40%, *p* = 0.0192962, Figure 7E), “Chromatin organization” (28.57%, *p* = 0.0208616, Figure 7F), and “Signal transduction” (11.46%, *p* = 0.0479025, Figure 7G). The two clusters “Reproduction” (*p* = 0.00029321) and “Circadian Clock” (*p* = 0.00294656) were enriched but not denoted as cancer-related. 

## 4. Discussion

Liver cancer is one of the most common cancers in China due to high numbers of infections with the hepatitis B virus, and the mechanism behind liver cancer is still insufficiently understood [36,37]. Hepatocellular carcinoma (HCC) covers the majority of cases of primary liver cancer. Previous research has indicated that USP15 plays a role in tumor development, but its regulatory function as an oncogene or tumor-suppressing gene has never been demonstrated. 

We investigated the function of USP15 in HCC through a synergetic application of experimental methods and system biology methods. In a publicly available dataset (LIHC, GEPIA2) of HCC patients, we found the level of USP15 expression reduced in tumor tissue and associated with high survival rate. Additionally, in our own cohort of 102 patients, we found expression of USP15 reduced in tumor tissue. Within our cohort, a high level of USP15 expression was associated with significantly decreased risk for mortality and cancer relapse. However, the mortality rate decreased sharply in the high-USP15 group at late HCC phase in the database and our clinical data. This would indicate that the tumor suppression role of USP15 only happened in the early phase of HCC, and in the end phase, the tumor suppression role disappeared. The results motivated a more detailed investigation of the function of USP15 and its potential role as a tumor-suppressing gene in liver cancer.

We investigated the phenotype of overexpression and knockout of USP15. Overexpression of USP15 in HCC cell lines correlated with slow proliferation and slow migration. Knockout of USP15 in HCC cell lines correlated with fast proliferation and fast migration. In a mouse model, overexpression of USP15 correlated with slow tumor growth. The experimental results suggested a preferable role of USP15 in HCC, presumably via a suppression of cell proliferation and motility.

To explore the relevant function of USP15, we constructed a PPI network. A set 105 interaction partners of USP15 contained two liver cancer genes, LRRK and NOTCH1. Ubiquitination of one of the 105 interaction partners may affect a larger set of 6175 proteins (layer 2 proteins). To illustrate, we selected eight layer 2 proteins that were known to be related to cancer. In a cell line, overexpression and knockout had effects on the expression of each of these eight proteins. Tumor-related proteins turned out to be regulated by USP15 via indirect interactions mediated by other proteins. Via indirect interactions, USP15 may be able to regulate a rather high number of proteins. From an investigation of a small set of only eight proteins, we were not able to identify the net effect on tumor progression. We found, for example, expressions of oncogene AKT and the tumor-suppressor gene Rb1 negatively correlated with the level of USP15 expression. A similar regulation of oncogene and tumor-suppressor gene may be interpreted as contradictory effects and their concerted effect on tumor progression is difficult to estimate.

Metastasis is a complex process that involves a cascade of multiple steps for the successful establishment of clinically impactful metastases. Metastasis involves multiple steps, including epithelial-mesenchymal transition (EMT), migration, matrix degradation, invasion into lymph vascular tissue, extravasation, adhesion, and mesenchymal-epithelial transition (MET) [20]. EMT involves disassembly of cell–cell junctions, actin cytoskeleton reorganization, and increased cell motility and invasion, as characterized by down-regulation and translocation of beta-catenin from the cell membrane to the nucleus and up-regulation of mesenchymal molecular markers such as vimentin, fibronectin and N-cadherin [38]. We found that N-cadherin was not regulated by USP15. USP15 may have no effect on EMT in metastasis. USP15 may, however, regulate a high number of genes that may be relevant for HCC. In our PPI network of USP15, we identified *n_HCC_* = 143 known liver cancer genes (mutation rate > 1%). To identify a large set of proteins that may be regulated by USP15, we applied NGS experiments and determined log-fold changes after a knockout of USP in a cell line (Ha22T). We found *n_diff_* = 478 proteins differentially expressed. 

The investigation of the regulatory effect of USP15 on single individual proteins may not lead to a conclusive picture of its role in HCC. We decided to explore further the function of USP15 not in terms of regulatory effects on individual proteins but in terms of regulatory effects on pathway hierarchies. We scored the 2443 human pathway hierarchies and selected 225 pathway hierarchies that were simultaneously relevant for liver cancer genes and differentially expressed proteins. Note that USP15 may contribute to multiple cellular functions, of which a majority may be irrelevant for tumor progression in HCC. We found functions “cell proliferation” and “cell migration” significantly enriched in the 225 pathway hierarchies that were predicted to be relevant for the role of USP15 in HCC. Note that the relevance of the functions “cell proliferation” and “cell migration” was already indicated by our cell line experiments. 

Compared to the gene regulation of individual proteins, pathway hierarchies may offer a higher level of viewpoint that can be helpful to understanding the regulation of tumorigenesis. We found the 255 relevant pathway hierarchies enriched in 8 of the 28 clusters, i.e., connected components, of the network of all human pathway hierarchies. Six of the clusters can easily be related to cancer, see Table 3 and Figure 7.

Prominent examples were the clusters for signal transduction and gene expression. The cluster signal transduction contained 69 out of 163 proliferation pathways and 35 out of 75 migration pathways, see Appendix A. The cluster gene expression contained 22 proliferation pathways. The enrichment of these two clusters indicated that USP15 may regulate the two clusters’ signal transduction and gene expression. By such a regulation, USP15 may influence cell proliferation and cell migration in tumorigenesis of HCC. 

The cluster cell cycle contains cell cycling checkpoints which play an important role in cancer [35]. The cluster DNA repair can be related to cancer progression because the accelerated replication of cancer cells induces so-called replication stress [39]. Replication stress in combination with an impaired DNA repair has been associated with cellular senescence and barriers to malignant progression [40,41]. The role of ubiquitination in DNA damage response has been previously reported by Ramadan and Dikic [17]. Peng et al. reported the involvement of USP15 in regulation of DNA-homologous recombination repair [18]. In particular, Peng et al. demonstrated the recruitment of USP15 to DNA double-strand breaks (DSBs) by the mediator of DNA damage checkpoint protein 1 (MDC1). Recruited to DSBs, USP15 can be deubiquitinated and demobilize a heterodimer of the BRCA1-associated RING domain protein 1 (BARD1) and the BRCA1 C Terminus (BRCT). Note that binding of heterodimer BARD1/BRCA1 to the damaged DNA prevents transcription and restores genetic stability [42]. The independent findings by other groups of a regulatory role of USP15 in DNA repair supports the meaningfulness of our approach.

The cluster cellular responses to external stimuli included pathways such as cellular response to heat stress, heat shock factor 1 (HSF1)-dependent transactivation, regulation of HSF1-mediated heat shock response, and HSF1 activation and attenuation phase, see Appendix A. HSF1 has been reported to function as a negative regulator of non-homologous end joining (NHEJ) in DNA repair activity [43]. Note that abnormal NHEJ plays an important role in tumorigenesis [44].

HSF1 has been reported to be involved in stress-induced cancer cell proliferation via IER5 [45]. 

The cluster cellular responses to external stimuli included additional pathways that are cancer-related, such as cellular senescence, DNA damage/telomere stress-induced senescence, and formation of senescence-associated heterochromatin foci, see Appendix A. Among the involved proteins were tumor protein P53 (TP53) and retinoblastoma protein (RB1), both of which may act as transcription factors or as a tumor suppressor. TP53 can bind to the promoter of cyclin-dependent kinase inhibitor 1 (CDKN1A) and may trigger transcription [46]. CDKN1A inhibits the activity of cell division protein kinase 2 (CDK2) and leads to an arrest of cell cycle at G1/S phase [47].

In chromatin organization, USP15-related cancer genes enriched in PKMTs and RMTs. 

Lysine methyltransferases (KMTs) and arginine methyltransferases (RMTs) regulate the Methylation levels of histone by adding methyl groups to histone, which play roles in regulating digestive cancer, including liver cancer [48]. H3K9me2 in the promoter region of RARRES3 gene suppresses its transcription and promotes cancer cell migration in HCC [49]. H3K27me3 enriched in the Kruppel-like factor 2 (KLF2) promoter leads to the tumor cell population in GC and CRC [50]. The decreased H3K27me3 in the promoter of SLUG activates the transcription and promotes migration and invasion in HCC [51]. All this evidence clearly showed that chromatin organization strongly relates to HCC and indicated that USP15 regulates HCC via histone methylation level.

To describe phenotypes, we found pathway hierarchies more suitable than a single gene or a small set of genes. Pathway clusters offered a coarser-grained—but still meaningful—viewpoint on regulation. Pathway clusters could be linked phenotypes and helped to understand the relations between USP15 and phenotypes of its regulatory function in HCC. 

## 5. Conclusions

We found high level of USP15 expression to be significantly correlated with high survival rate in HCC patients. Relevant cellular phenotypes were slow proliferation and slow migration. We confirmed the role of USP15 as a tumor suppressor in a mouse model. To link the experimentally observed phenotypes to molecular function of USP15 we constructed a protein–protein interaction network. We found the regulatory function of USP15 to be mediated by indirect interactions. Indirect interactions may enable USP15 to regulate thousands of other proteins. The phenotype of complex regulation is hard to predict. We proposed and tested a theoretical approach based on pathway hierarchies. The viewpoint of pathway hierarchies offered a meaningful coarse-grained viewpoint on the regulatory function of USP15. We were able to link our experimentally observed phenotypes to clusters of pathways. Furthermore, in other applications, clusters of pathways may be an appropriate viewpoint to understand the regulatory function of proteins and may offer a starting point to select molecular mechanisms for further experimental investigation. 

## Figures and Tables

**Figure 2 cancers-15-01371-f002:**
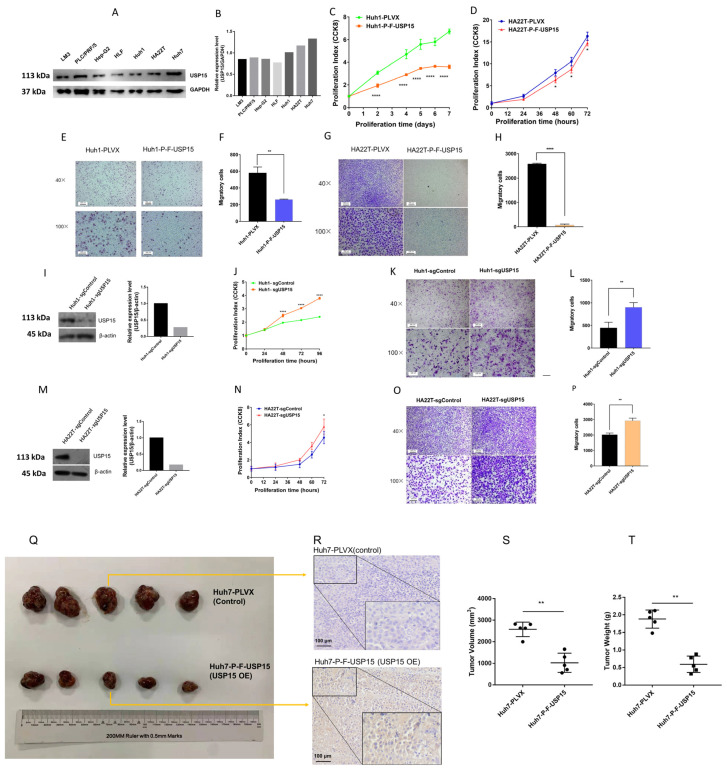
The expression of USP15 is increased in hepatocellular carcinoma cell lines. Overexpression of USP15 correlates with slow proliferation, slow migration, and slow tumor growth. Knockout of USP15 correlates with fast proliferation and fast migration. (**A**) Western blot of USP15 protein expression in cell lines LM3, PLC/PRF/5, HLF, Hep-G2, Huh1, HA22T, and Huh7. Glyceraldehyde 3-phosphate dehydrogenase (GAPGH) is the reference. (**B**) Bar chart of the normalized expression of USP15. We applied GAPGH to normalize the intensity values. HCC cell lines Huh1, HA22T, and Huh7 have the highest expression levels of USP15. (**C**) The proliferation index (CCK8 assay) of transfected cell line Huh1-P-F-USP15 (red line) versus control Huh1-PLVX (green line), see text. Overexpressing cells, Huh1-P-F-USP15, had a decreased proliferation index (*p* values at 5 time points all less than 0.0001, two-tailed *t*-test). (**D**) The proliferation index (CCK8 assay) of transfected cells line HA22T-P-F-USP15 (red line) versus control HA22T-PLVX (green line), see text. Overexpressing cells, HA22T-P-F-USP15, had a decreased proliferation index (*p* values at 48 h, 60 h and 72 h are less than 0.05, two-tailed *t*-test). (**E**) Images of migration assays (Transwell) for cell lines Huh1-PLVX (left images) and Huh1-P-F-USP15 (right images). Images are shown in two magnifications, 40× (top images) and 100× (bottom images). (**F**) Counts of migratory cells for transfected cell line Huh1=P-F-USP15 (right bar) versus control Huh1-PLVX (left bar), see text. Overexpressing cells, Huh1=P-F-USP15, had a decreased number of migratory cells (*p* < 0.01, two-tailed *t*-test). (**G**) Images of migration assays (Transwell) for cell lines HA22T-PLVX (left images) and HA22T-P-F-USP15 (right images). Images are shown in two magnifications, 40× (top images) and 100× (bottom images). (**H**) Counts of migratory cells for transfected cell line HA22T-P-F-USP15 (right bar) versus control HA22T-PLVX (left bar), see text. Overexpressing cells, HA22T-P-F-USP15, had a decreased number of migratory cells (*p* < 0.0001, two-tailed *t*-test). (**I**) Western blot of USP15 protein expression in control cell line, Huh1-sgControl, and knockout cell line, Huh1-sgUSP15, see text. Knockout of USP15 expression by CRISPR/Cas9 is visible for Huh1-sgUSP15 versus control, Huh1-sgControl. (**J**) The proliferation index (CCK8 assay) of control cell line Huh1-sgControl (green line) and knockout cell line Huh1-sgUSP15 (red line), see text. Knockout of USP15 expression in cells, Huh1-sgUSP15, led to an increased proliferation index (*p* value at 48 h, 72 h and 96 h are less than 0.0001, two-tailed *t*-test). (**K**) Images of migration assays (Transwell) for knockout cell line Huh1-sgUSP15 (right images) and control cell lineHuh1-sgControl (left images). Images are shown in two magnifications, 40× (top images) and 100× (bottom images). (**L**) Counts of migratory cells for knock cell line Huh1-sgUSP15 (right bar) versus control Huh1-sgControl (left bar), see text. Knockout cells, Huh1-sgUSP15, had an increased number of migratory cells (*p* < 0.01, two-tailed *t*-test). (**M**) Western blot of USP15 protein expression in control cell line, HA22T-sgControl, and knockout cell line, HA22T-sgUSP15, see text. Effect of knockout of USP15 expression by CRISPR/Cas9 is visible for HA22T-sgUSP15 versus control, Huh1-sgControl. (**N**) The proliferation index (CCK8 assay) of control cell line HA22T-sgControl (green line) and knockout cell line HA22T-sgUSP15 (red line), see text. Knockout of USP15 expression in cells, HA22T-sgUSP15, led to an increased proliferation index (*p* value at 72 h less than 0.05, two-tailed *t*-test). (**O**) Images of migration assays (Transwell) for knockout cell line HA22T-sgUSP15 (right images) and control cell line HA22T-sgControl (left images). Images are shown in two magnifications, 40× (top images) and 100× (bottom images). (**P**) Counts of migratory cells for knock cell line HA22T-sgUSP15 (right bar) versus control HA22T-sgControl (left bar), see text. Knockout cells, HA22T-sgUSP15, had an increased number of migratory cells (*p* < 0.01, two-tailed *t*-test). (**Q**) Images of tumor growth in an animal model of nude mice. A control group, upper row, was injected by control cells, Huh7-PLVX. A second group, lower row, was injected by transfected cells, Huh7-P-F-USP15, that overexpressed USP15. Tumors were excised 25 days after injection. (**R**) Two exemplary images of immunochemically staining (dye: 3,3′-diaminobenzidine, DAB) of tumor tissue of nude mice. Upper image shows low yellow staining for USP15 of control cells, Huh7-PLVX. Lower image shows strong yellow staining for USP15 of transfected cells, Huh7-P-F-USP15. H&E counterstaining colors cell nuclear in blue-purple. (**S**) Box plots of tumor volumes for two HCC cell lines, Huh7-PLVX (control) and Huh7-P-F-USP15 (transfected). Tumors of transfected cells, Huh7-P-F-USP15 (lower row), have smaller dimensions than tumors of the control group, Huh7-PLVX (*p* < 0.01, two-tailed *t*-test). OE—overexpressing. (**T**) Box plots of tumor weights for two HCC cell lines, Huh7-PLVX (control) and Huh7-P-F-USP15 (transfected). Tumors of transfected cells, Huh7-P-F-USP15 (lower row), have less weight than tumors of the control group, Huh7-PLVX. *: *p* < 0.05; **: *p* < 0.01; *****: p* < 0.0001.

**Figure 3 cancers-15-01371-f003:**
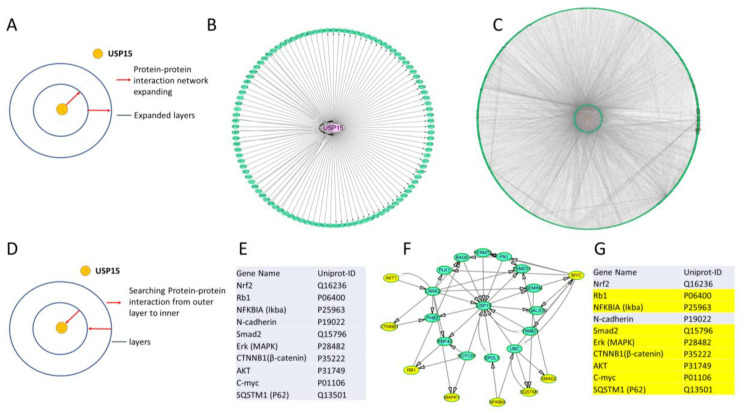
Human USP15 PPI network, a network of proteins that interact directly or indirectly with USP15. The role of eight experimentally accessible proteins. (**A**) Layer 1 proteins interact with USP15, and layer 2 proteins interact with the layer 1 proteins. (**B**) Visualization (circle layout, Cytoscape) of interactions of 105 proteins of layer 1 (outer circle) with USP15 (center node). (**C**) Visualization (circle layout, Cytoscape) of indirect interaction of 6175 proteins of layer 2 (outer circle) with USP15 (center). (**D**) Proteins of layer 2 are linked to USP15 via a protein of layer 1 (inner circle). (**E**) Choice of ten proteins that are tumor-related and experimentally accessible. (**F**) Visualization (circle layout, Cytoscape) of indirect interaction of eight preselected proteins (highlighted in yellow, outer circle) with USP15 (center node). (**G**) Two of the ten preselected proteins were not members of the Human USP15 PPI network (not highlighted in yellow).

**Figure 4 cancers-15-01371-f004:**
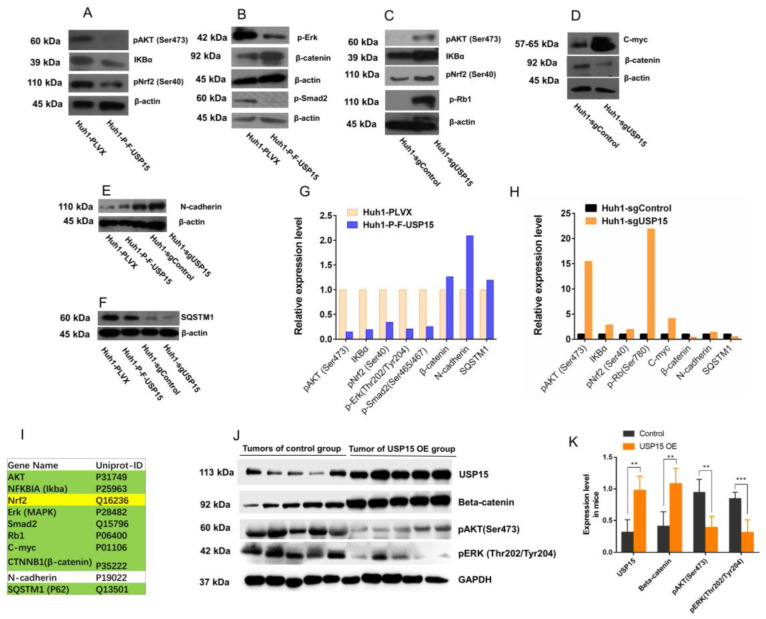
Tumor-related proteins can be regulated by USP15 via interactions mediated by other proteins. (**A**) Western blots of protein IKBα and phosphorylated proteins p-Nrf2 and p-AKT in cell lines Huh1-P-F_USP15 (overexpressing USP15) and Huh1-PLVX (control). (**B**) Western blots of protein β-catenin and phosphorylated proteins, p-Smad2 and p-Erk in cell lines Huh1-P-F_USP15 (overexpressing USP15) and Huh1-PLVX (control). (**C**) Western blots of proteins IKBα and phosphorylated proteins p-Nrf2, p-Rb1, and p-AKT in cell lines Huh1-sgUSP15 (knockout USP15) and Huh1-sgControl (control). (**D**) Western blots of proteins C-myc and β-catenin in cell lines Huh1-sgUSP15 (knockout USP15) and Huh1-sgControl (control). (**E**) Western blots of proteins N-cadherin in cell lines Huh1-P-F_USP15 (overexpressing USP15) and Huh1-sgUSP15 (knockout USP15) versus control. (**F**) Western blots of proteins SQSTM1 in cell lines Huh1-P-F_USP15 (overexpressing USP15) and Huh1-sgUSP15 (knockout USP15) versus control. (**G**) Bar plot of intensities, normalized to ß-actin, with (Huh1-P-F_USP15) and without (Huh1-PLVX) overexpression of USP15. Only differentially expressed proteins are shown. (**H**) Bar plot of intensities, normalized to ß-actin, with (Huh1-sgUSP15) and without (Huh1-sgControl) knockout of USP15. Only differentially expressed proteins are shown. (**I**) List of investigated proteins. Interaction partners of USP in the human USP PPI network are regulated by USP15 and highlighted in green. Nrf2 and N-caderin are not members of the human USP PPI network. Whereas Nrf2 (highlighted yellow) is regulated by USP15, N-caderin is not. (**J**) Western blots of downstream proteins of USP15 in tumor samples. The left side is the tumor of HCC cell lines, Huh7-PLVX, which is used as control. The right side is the tumor of Huh7-P-F-USP15 (**K**). Band densitometry intensity ration of the Western blot in picture (**J**). **: *p* < 0.01; ***: *p* < 0.001.

**Figure 5 cancers-15-01371-f005:**
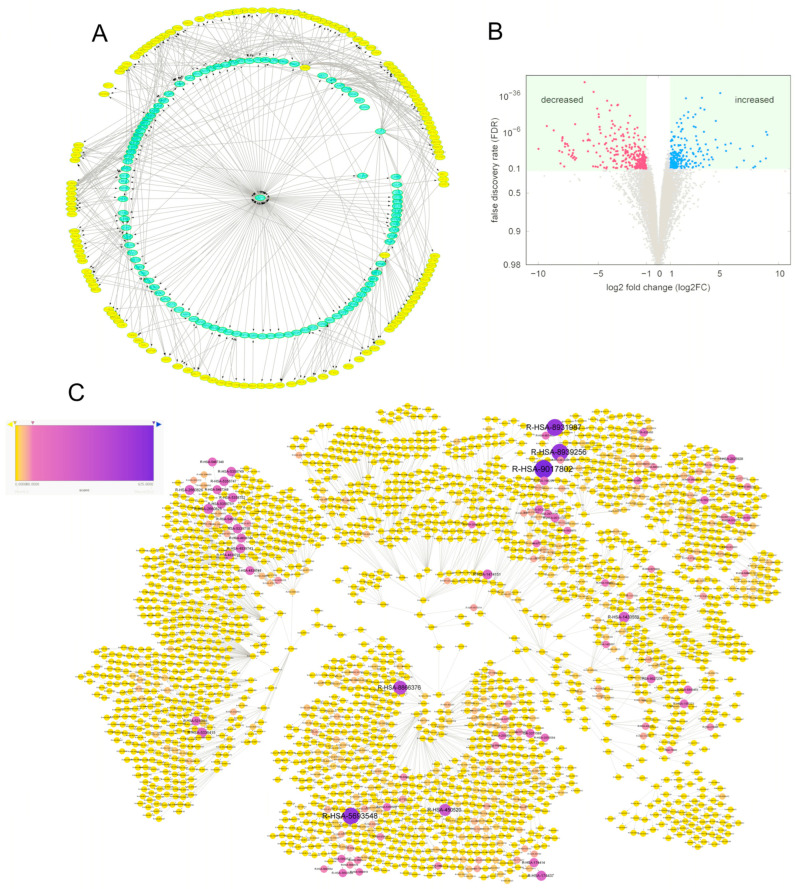
Regulatory role of USP15 in HCC: pathway hierarchies. (**A**) Sub-network of the human USP15 PPI network. A total of 143 liver cancer genes (highlighted in yellow) interact directly or indirectly with USP15. (**B**) Volcano map of the effect of knockout of USP15 in cell line HAT22T. The log-fold change of 14,099 identified genes in cell line HA22T sgUSP15 (knockout USP15) versus cell line HA22T sg-control as control, see material and methods. A fraction of 1.4% (203 proteins, highlighted in blue) showed increased expression levels (false discovery rate < 0.1, log-fold change > 1). A fraction of 2.0% (278 proteins, highlighted in red) showed decreased expression (false discovery rate < 0.1, log-fold change < −1). (**C**). Network of all human pathway hierarchies. Nodes with high score s are highlighted in dark blue and may contribute to the regulatory role of USP15 in HCC, see text.

**Figure 6 cancers-15-01371-f006:**
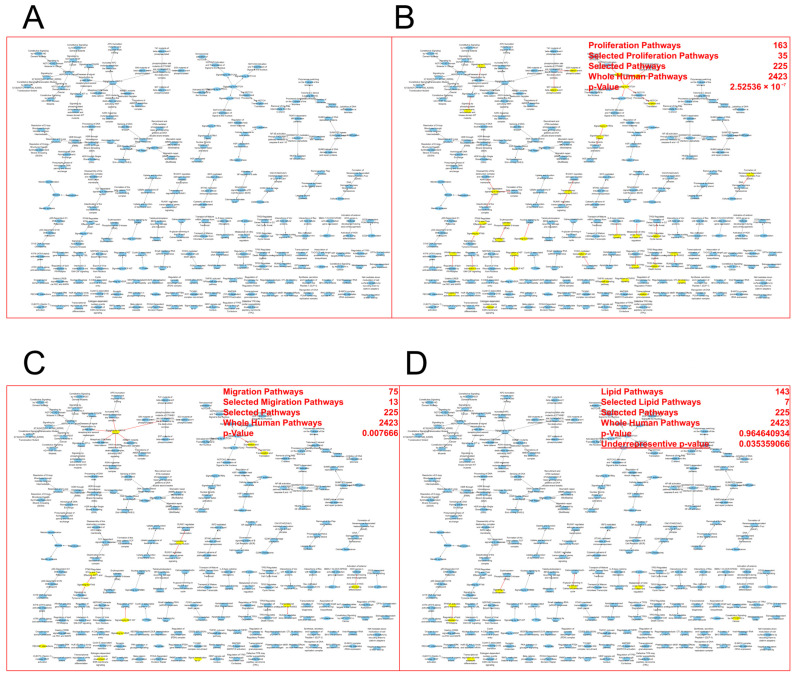
USP15 may repress HCC via genes in cell proliferation and cell migration. HCC-USP15 pathways: network of pathway hierarchies that may contribute to the regulatory role of USP15 in HCC. (**A**) A sub-network of the human pathway hierarchies. Each node has a score s > 20 and hence may contribute to the regulatory role of USP15 in HCC, see text. (**B**) Pathway hierarchies associated to the function “cell proliferation” are highlighted in yellow. The function “cell proliferation” is significantly enriched in the network (*p* =2.5 × 10^−7^, hypergeometric test). (**C**) Pathway hierarchies associated to the function “cell migration” are highlighted in yellow. The function “cell migration” is significantly enriched in the network (*p* = 0.0077, hypergeometric test). (**D**) Pathway hierarchies associated to the function “lipid processes” are highlighted in yellow. The function “lipid processes” is significantly underrepresented in the network (*p* = 0.035359066, hypergeometric test).

**Figure 7 cancers-15-01371-f007:**
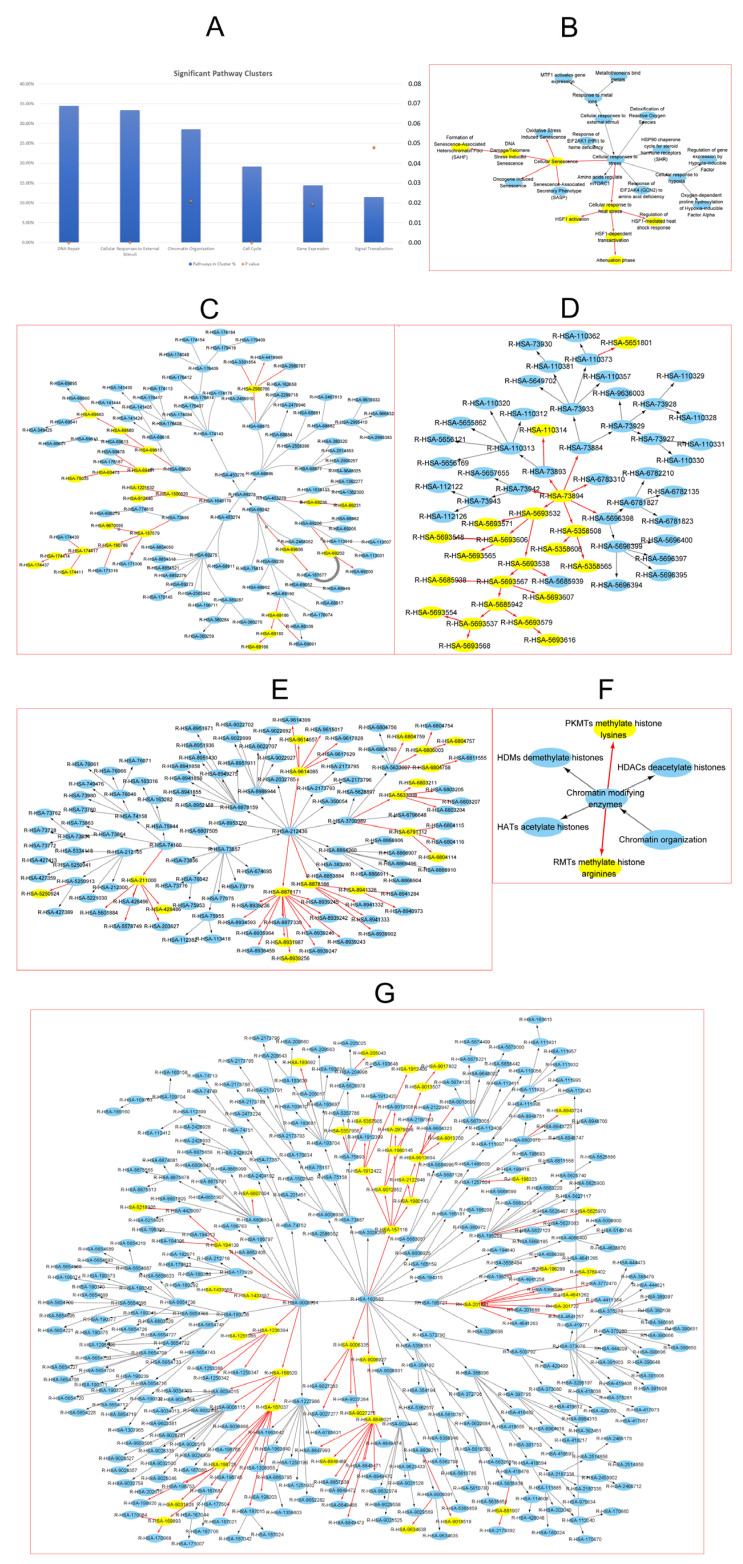
USP15 may repress HCC via six pathway clusters. USP15-HCC pathways are significantly enriched in six cancer-related clusters of the complete network of human pathway hierarchies. (**A**) Degree of enrichment (as a percentage) of the six cancer-related clusters. (**B**) Cluster “Cellular responses to external stimuli”. (**C**) Cluster “Cell Cycle”. (**D**) Cluster “DNA repair”. (**E**) Cluster “Gene expression”. (**F**) Cluster “Chromatin organization”. (**G**) Cluster “Signal transduction”. UP15-HCC pathways are highlighted in yellow.

**Table 1 cancers-15-01371-t001:** Clinical and pathological characteristics of the patient cohort of 102 patients suffering from HCC. A tissue sample of each patient was immunohistochemically stained (IHC) for USP15. By visual inspection, two pathologists classified the stained tissue into low or high expression, see text. All *p*-values (two-tail Fisher’s exact test [22]) are above 0.05, and hence, no significance differences of USP15 expression exist in the patient subgroups of clinical characteristics, e.g., age (young/old) and gender (male/female). Alpha-fetoprotein—AFP; hepatitis B surface antigen—HBsAg.

	Total	USP15 Expression		*p*-Value
		Low (%)	High (%)	
Age				
<53	47	37.25	8.83	0.254
≥53	55	37.25	16.67	
Gender				
Male	85	63.73	19.61	0.363
Female	17	10.78	5.88	
AFP (ng/mL)				
<400	63	45.10	16.67	0.816
≥400	39	29.41	8.82	
HBsAg				
Negative	16	12.75	2.94	0.755
Positive	86	61.76	22.55	
Tumor size				
<5 cm	61	44.12	15.69	1.0
<5 cm	41	30.39	9.80	
≥5 cm				
Tumor number				
Single	89	63.73	23.53	0.507
Multiple	13	10.78	1.96	
Liver cirrhosis				
No	45	29.41	14.71	0.116
Yes	57	45.10	10.78	

**Table 2 cancers-15-01371-t002:** Sequence of primers. F—forward; R—reverse.

Name	Sequences (5′–3′)
SgRNA-1	F: 5′-CACCGCGTCGCGATGTCAGACCGC-3′
SgRNA-1	R: 5′-AAACGCGGTCTGACATCGCGACGC-3′
EcoRI-USP15	F: 5′-ATTACGCTGAATTCATGGCGGAAGGCGGAGCGGCGGAT-3′
BamHI-USP15	R: 5′-CGACTCTAGAGGATCCTA TTAGTTAGTGTGCATACAGT-3′

**Table 3 cancers-15-01371-t003:** The 28 connected components of the complete network of human pathway hierarchies. HCC-USP15 pathways were significantly enriched in eight connected components, so-called clusters. We denoted six of the enriched clusters (in bold) as relevant to tumor research, highlighted in boldface. Two clusters, “Reproduction” and “Circadian Clock” (in italic), were significantly enriched but not denoted as cancer-related. #: numbers.

Cluster	#Nodes
Hemostasis	39
Intrinsic pathway for Apoptosis	20
Neuronal System	78
Developmental Biology	59
Metabolism	325
*Reproduction*	8
Extracellular matrix organization	19
Cell–Cell communication	13
**Signal Transduction**	382
**Cell Cycle**	122
Disease	517
Immune System	197
Organelle biogenesis and maintenance	13
Transport of small molecules	69
Metabolism of proteins	124
Muscle contraction	11
*Circadian Clock*	4
**Chromatin organization**	7
Programmed Cell Death	21
Vesicle-mediated transport	38
DNA Replication	19
**DNA Repair**	61
**Gene expression (Transcription)**	125
Metabolism of RNA	51
**Cellular responses to external stimuli**	24
Digestion and absorption	7
Protein localization	6
Autophagy	10

## Data Availability

Raw figures were uploaded as manuscript submission. NGS dataset were dropped here: https://figshare.com/articles/dataset/NGS/17075690.

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
