# Peer review of "USP15 Represses Hepatocellular Carcinoma Progression by Regulation of Pathways of Cell Proliferation and Cell Migration: A System Biology Analysis"

_cancers, 2023, doi:10.3390/cancers15051371_

Round 1

Reviewer 1 Report (Previous Reviewer 1)

The authors have successfully addressed all my previous minor concerns. Therefore, it is my opinion that the manuscript can be published now in its present form.

Reviewer 2 Report (Previous Reviewer 2)

I am pleased to see that the manuscript has considerably improved compared to the previous version. The authors answered all the critiques with convincing responses. The revised manuscript content, quality of presentation and various changes incorporated will definitely promote interest to the scientific community. Overall, I am satisfied with the revised manuscript.  

This manuscript is a resubmission of an earlier submission. The following is a list of the peer review reports and author responses from that submission.

Round 1

Reviewer 1 Report

In this study by Ren et al, the authors explore the role of USP15 as a tumor suppressor in hepatocellular carcinoma. The authors have used a wide range of techniques over their own cohort of patients, and also data available in public databases. They have evaluated gene expression, protein production, in vitro and in vivo growth of many tumor cell lines, etc, which all of them support their hypothesis. They did an extensive systems biology study to evaluate the role of this ubiquitin kinase in intracellular pathways.

It is my opinion that this paper merits publication, pending to solve some minor issues, but which I think are important.

1. Overall survival, Figure 1B. It is evident that the high USP15 group has increased overall survival. However, there is a curious crossover of the curves at 80 months, whereby the group with low USP15 ends up with a slightly increased proportion of survivers compared to the high group. This is curious, and it would be important for the authors to comment on this. What is the cause of this crossover? 

2. Something similar with figure 1E. Although the high group has clearly increased overall survival, then suddenly nearly at the same time, all patients die (50 months). It would be interesting for the authors to provide some comments on this. Has this cohort been selected as "fast progressors" as the authors comment on the text?

3. The figures with systems biology data. I am aware of the difficulty in representing these figures, as I have also some experience with them. The problem is that the names of the genes are often so small that it is nearly impossible to read them, unless the figure is of extremely high resolution. I would advise the authors to highlight some of the key regulators in gene pathways in their figures, by selecting a few and labelling them specifically with a bigger letter size. 

4. I see that supplementary figures have been included in the text together with the main figures. I would suggest the authors to label these supplementary figures as main figures. Cancers allows it, and it would be nice to have all figures in the text as main figures. No need to label them as Figures S...

Reviewer 2 Report

The authors evaluated the function of USP15 as an anti-tumor gene in hepatocellular carcinoma (HCC) progression. Using a System Biology method of analysis, the authors identified the interactome and behavioral components of USP15. The authors used information from databases and demonstrated the importance of USP15 in HCC tissues and directly co-related the gene expression with patient survival. Various HCC cell lines were used to analyze the expression of USP15 and its relevance on cell proliferation and cell migration. To show the in-vivo effect of USP15, the authors used mice models and showed tumor regression when subcutaneously injected with Huh7-P-F-USP15. Next, the authors used database studies and found the protein-protein interacting partners of USP15 and validated some of the findings. To co-relate the interrelation of USP15 with other network/pathways the authors used various databases and found interesting relationship between USP15 and HCC proteins.  Various genes were also identified that differentially expressed as seen from the volcano plot (NGS) with respect to USP15 expressions. These experiments were followed by more studies on networks and classified according to the related pathways. Accordingly, the authors could justify the title of the manuscript that USP15 functions as anti-tumor mostly by targeting cell proliferation and migration. Here are my comments.

1.       Figure S1. Please move the images from the main manuscript to a supplementary figure file instead.  Even with the magnified figures it is very difficult to see the cytoplasmic or membrane staining of USP15. Please provide a much more magnified image.

2.       Figure 1A, mention how many tumor and non-tumor samples were analyzed.

3.       Figure 1C, it is not mentioned neither in the result section nor in the figure legend which one of the two samples is tumor and non-tumor. Please provide more magnified images of the same.

4.       Figure S2, if this is a supplementary figure, please put it separately in a supplementary file instead of the main manuscript figure. There is no scale bar in the figure. Also it is not mentioned which cell migration picture represents what time points. What time point does the right panel of the cell migration images represents. I believe the first one is 0 hour time point.

5.       Figure S3, same comments as above comment. No scale bar and no representation of time points of the cell migration images.

6.       Figure 2A, please include molecular weight marker in the blot

7.       For Figure 2C and 2D please show the overexpression efficiency of the plasmids for PF-USP15 and compare it to the control vector PLVX  in both cell lines Huh1 and HA22T for validation.

8.       Figure 2C and 2D. Please include Ki67 confocal images for the cell proliferation assay. Also mention if the slow rate of cell proliferation is due to apoptosis or arrest in the cell cycle.

9.       Figure 2E. Please include scale bar for the images. Please show the section of the image in the 40X that is magnified to 100x or are these different images.

10.   Figure 2G. same as comment number 9. Please include scale bar.

11.   Figure 2I, please include molecular weight marker in the blot

12.   Figure 2K, please include scale bar.

13.   Figure 2M, please include molecular weight marker in the blot

14.   Figure 2O, please include scale bar.

15.   Figure  2R, please include a magnified image. Also please perform a western blot to show the levels of USP15 comparing the control and USP15 overexpressed tissues from the tumor samples for validation. Also include some downstream target genes for USP15 to show the decrease in tumorigenicity.

16.   Figure 2S. please include the tumor weight

17.   Figure 4. Here the authors would like to perform the validation of USP15 interaction with other candidate genes. However, the experiments suggest otherwise. To validate the interaction of USP15 with other genes an immuno-precipitation experiment (protein-protein interaction) study by western blot is needed to be shown instead of the analyzing the gene expression levels. Figure 4A-4F shows the regulation of these genes by USP15 which can be direct or indirectly regulated and it does not prove that there would be any protein-protein interactions involved. To show the validation of interaction please perform an immuno-precipitation experiment of USP15 with the other genes.

18.   Figure 4. Please include molecular weight marker in all the blots and use a single color black for all the text in the blots. Please remove the outer border line in red.

19.   Figure 5B, mention how many samples were analyzed for the NGS experiment.

20.   Figure 5A and 5C. it is very difficult to interpret the figure

21.   Figure S4, do the authors plan to use this as supplementary figure or as a main figure.

22.   Figure 6A – D, it is very difficult to interpret anything from the figures

23.   Figure 7B-7E and 7G it is very difficult to interpret anything from the figures